# Inverse Parameter Identification for Hyperelastic Model of a Polyurea

**DOI:** 10.3390/polym13142253

**Published:** 2021-07-09

**Authors:** Yihua Xiao, Ziqiang Tang, Xiangfu Hong

**Affiliations:** 1School of Mechatronics and Vehicle Engineering, East China Jiaotong University, Nanchang 330013, China; 2State Key Laboratory of Performance Monitoring and Protecting of Rail Transit Infrastructure, East China Jiaotong University, Nanchang 330013, China; TZQ1174707349@163.com (Z.T.); hxiangfu@163.com (X.H.)

**Keywords:** polyurea, hyperelastic model, inverse procedure, finite element, experiment

## Abstract

An inverse procedure was proposed to identify the material parameters of polyurea materials. In this procedure, a polynomial hyperelastic model was chosen as the constitutive model. Both uniaxial tension and compression tests were performed for a polyurea. An iterative inverse method was presented to identify parameters for the tensile performance of the polyurea. This method adjusts parameters iteratively to achieve a good agreement between tensile forces from the tension test and its finite element (FE) model. A response surface-based inverse method was presented to identify parameters for the compression performance of the polyurea. This method constructs a radial basis function (RBF)-based response surface model for the error between compressive forces from the compression test and its FE model, and it employs the genetic algorithm to minimize the error. With the use of the two inverse methods, two sets of parameters were obtained. Then, a complete identified uniaxial stress–strain curve for both tensile and compressive deformations was obtained with the two sets of parameters. Fitting this curve with the constitutive equation gave the final material parameters. The present inverse procedure can simplify experimental configurations and consider effects of friction in compression tests. Moreover, it produces material parameters that can appropriately characterize both tensile and compressive behaviors of the polyurea.

## 1. Introduction

Polymers such as engineering plastics and rubbers have been widely applied in industry. Their mechanical properties have attracted extensive research interests. Many researchers have studied the tension performance [1], compression performance [2,3,4], strain-rate-dependent behavior [4,5,6], and impact resistance of polymers [7]. Polyurea is a relatively new polymer material. It possesses many attractive properties, such as excellent ductility, good adhesion, high wear resistance, and high impact resistance. It has been increasingly used in the water proofing, anti-corrosion, and impact protection of engineering structures. To design engineering structures with polyurea materials, it is of great significance to have accurate constitutive models and corresponding model parameters for the structural reasonability and reliability.

The investigation of mechanical behaviors and constitutive models of polyurea materials has attracted extensive research interests. Raman et al. [8] performed an experimental study on the tensile behavior of polyurea with the strain rate ranging from 0.006 to 388 s^−1^. Mohotti et al. [9] proposed a strain-rate-dependent constitutive model to predict the high-strain-rate behavior of polyurea under tensile loading. Bai et al. [10] established a hyper-viscoelastic constitutive model to characterize the material behavior of polyurea subjected to uniaxial compressive loading. Guo et al. [11] developed a visco-hyperelastic constitutive model to describe the mechanical behavior of a polyurea material and validated its effectiveness for different loading conditions and strain rates. Li and Lua [12] developed a constitutive model considering both hyperelastic and viscoelastic behaviors of polyurea. Gamonpilas and McCuiston [13] proposed a nonlinear viscoelastic constitutive model for polyurea.

In previous studies, material parameters in constitutive models of polyurea have generally been obtained by directly fitting stress–strain curves derived from standard tests, such as the uniaxial tension test and compression test. During the derivation of stress–strain curves from standard test results, it is required to assume simple stress states. For example, the assumption of one-dimensional stress state is required for uniaxial tension and compression tests. However, stress states for practical uniaxial tension and compression tests usually deviate from the ideal one-dimensional stress state to some extent. Especially, for uniaxial compression tests, friction between the specimen and test machine is inevitable and results in a nonuniform stress distribution in a specimen. Thus, the direct fitting method brings errors to material parameters. Besides, in most previous studies, material parameters for polyurea have been determined by either a tension or compression test. They may be only applicable to special loading conditions, for example, tension or compression loading only. This phenomenon is highlighted in the latter part of this work.

The inverse identification technique provides an alternative for overcoming the problem in the direct derivation of material parameters with standard tests. Many inverse identification methods have been developed to identify mechanical parameters for various materials [14,15,16,17,18,19,20,21]. The general idea of inverse parameter identification is to perform experimental tests, establish corresponding numerical models, and optimize material parameters to minimize the mismatch between experimental and numerical results. An obvious advantage of inverse parameter identification is that it allows complex stress states in experimental tests and can simplify experimental procedures. In this work, an iterative inverse method and a response surface-based inverse method are presented to identify hyperelastic model parameters for the tensile and compressive performance of polyurea, respectively. Based on this, the parameters that describe both tensile and compressive performance are determined.

The remainder of the paper is organized as follows: Section 2 briefs the constitutive model used for polyurea materials; Section 3 details the identification of material parameters for the tensile performance of a polyurea; Section 4 details the identification of material parameters for the compressive performance of the polyurea; Section 5 presents the determination and validation of material parameters that can appropriately characterize both tensile and compressive performance; the last section draws the main conclusions of this work.

## 2. Constitutive Model for Polyurea

Polyurea is generally considered an isotropic incompressible hyperelastic material. The polyurea considered in this work was the one-part polyurea SWD 9526, which was provided by SWD New Material (Shanghai) Co., Ltd. (Shanghai, China) It is mainly composed of 4,4′-diphenylmethane diisocyanate, polyether polyol, and silicon dioxide. It has the characteristics of a hyperelastic material, according to our material tests detailed below. Its deformation appeared to be elastic during a large strain range, and its stress–strain relationship showed apparent nonlinearity. A hyperelastic constitutive model was employed to characterize its material property. It is known that there are many hyperelastic constitutive models, such as the Mooney–Rivlin model, Ogden model, and polynomial model. In these models, different strain energy functions are used to characterize the stress–strain relationship. The polynomial model was adopted here, as the order of the polynomial strain energy function of this model can be adjusted flexibly to characterize material properties accurately. The general form of the strain energy function for the polynomial model is defined as
(1)$$U=\sum_{i+j=1}^{N}C_{ij}{\left({\bar{I}}_{1}-3\right)}^{i}{\left({\bar{I}}_{2}-3\right)}^{j}+\sum_{i=1}^{N}\frac{1}{D_{i}}{\left(J-1\right)}^{2i}$$
where *U* is the strain energy per unit volume, *N* is the order of the polynomial model, *C_ij_* and *D_i_* are material constants, *J* is the volume ratio, and ${\bar{I}}_{1}$ and ${\bar{I}}_{2}$ are defined as
(2)$${\bar{I}}_{1}={\bar{\lambda }}_{1}^{2}+{\bar{\lambda }}_{2}^{2}+{\bar{\lambda }}_{3}^{2}$$
and
(3)$${\bar{I}}_{2}={\bar{\lambda }}_{1}^{-2}+{\bar{\lambda }}_{2}^{-2}+{\bar{\lambda }}_{3}^{-2},$$
respectively, where ${\bar{\lambda }}_{i}=J^{-1/3}\lambda_{i}\;\left(i=1,2,3\right)$, and *λ**_i_* is the principal stretch ratio. For an incompressible material, $J=\lambda_{1}\lambda_{2}\lambda_{3}=1$. Thus, Equation (1) can be rewritten as
(4)$$U=\sum_{i+j=1}^{N}C_{ij}{\left(I_{1}-3\right)}^{i}{\left(I_{2}-3\right)}^{j},$$
where
(5)$$I_{1}=\lambda_{1}^{2}+\lambda_{2}^{2}+\lambda_{3}^{2},$$
and
(6)$$I_{2}=\lambda_{1}^{2}\lambda_{2}^{2}+\lambda_{1}^{2}\lambda_{3}^{2}+\lambda_{2}^{2}\lambda_{3}^{2}.$$

During the following inverse parameter identification, the order of the polynomial model is taken as 2 in order to appropriately describe material behaviors of the polyurea and limit the number of material parameters. The second-order strain energy function is
(7)$$U=C_{10}\left(I_{1}-3\right)+C_{01}\left(I_{2}-3\right)+C_{20}{\left(I_{1}-3\right)}^{2}+C_{11}\left(I_{1}-3\right)\left(I_{2}-3\right)+C_{02}{\left(I_{2}-3\right)}^{2},$$
where *C*_10_, *C*_01_, *C*_20_, *C*_11_, and *C*_02_ are material parameters that need to be identified.

To identify the five material parameters, direct experimental methods usually employ uniaxial tension or/and compression tests, where specimens are under quasi-uniaxial tension or/and compression states. For an ideal uniaxial stress state, the three principal stretch ratios are
(8)$$\lambda_{1}=\lambda_{U},\;\lambda_{2}=\lambda_{3}=\lambda_{U}^{-1/2},$$
where *λ*_U_ is the principal stretch ratio in the loading direction, and the two strain invariants can be expressed as
(9)$$I_{1}=\lambda_{U}^{2}+\frac{2}{\lambda_{U}},\;I_{2}=2\lambda_{U}+\frac{1}{\lambda_{U}^{2}}.$$

With the use of the principle of virtual work, the nominal stress *T*_U_ in the loading direction can be expressed as
(10)$$T_{U}=\frac{\partial U\;}{\partial \lambda_{U}}=\frac{\partial U\;}{\partial I_{1}}\frac{\partial I_{1}\;}{\partial \lambda_{U}}+\frac{\partial U\;}{\partial I_{2}}\frac{\partial I_{2}\;}{\partial \lambda_{U}}.$$

Inserting Equations (7) and (9) into Equation (10) yields
(11)$$T_{U}=2\left(1-\frac{1}{\lambda_{U}^{3}}\right)\left\{C_{10}\lambda_{U}+C_{01}+2C_{20}\lambda_{U}\left(I_{1}-3\right)+C_{11}\left[\left(I_{1}-3\right)+\lambda_{U}\left(I_{2}-3\right)\right]+2C_{02}\left(I_{2}-3\right)\right\}.$$

The above equation defines the *T*_U_–*λ*_U_ relationship for an ideal uniaxial stress state, which is a linear equation of the five material parameters. Once the *T*_U_–*λ*_U_ relationship is tested with experiments, the five material parameters can be directly determined by fitting the test data with Equation (11). However, it is not easy to obtain an exact *T*_U_–*λ*_U_ relationship via experimental methods for compressive loading, because in traditional uniaxial compression tests, friction inevitably exists between the test machine and specimen and results in a deviation from the ideal uniaxial stress state. For tensile loading, it is relatively easy to obtain a good test result of the *T*_U_–*λ*_U_ relationship, compared with compression loading, but this should be achieved with the aid of a special deformation meter tracking the deformation of the gauge length of a tension specimen.

This work introduces inverse methods to identify the five material parameters for the tensile and compressive performance of the polyurea. They do not require a special deformation meter to track the deformation of the gauge length of a specimen in a tension test, and they allow friction and the complex stress state in a compression test. The methods can simplify experimental procedures and produce accurate material parameters.

## 3. Inverse Identification of Material Parameters for Tensile Performance

### 3.1. Uniaxial Tension Test

Uniaxial tension tests were performed to provide experimental data to inversely identify material parameters for the tensile performance of the polyurea. The specimens used in the tests were dumbbell-shaped, as shown in Figure 1a. The nominal dimensions of the specimens are given in Table 1. The specimens were fabricated by the casting method. To fabricate a specimen, polyurea is poured into a lower mold (see Figure 1b). After the lower mold is filled up with polyurea, an upper mold is pressed in the lower mold. The sample is left in the mold to cure for one week. Then, it is taken out of the mold and polished with 2000-mesh abrasive papers. The specimen fabricated is shown in Figure 1c.

The uniaxial tension tests were carried out with a universal testing machine produced by Dongguan Deray Instrument Co., Ltd. (Dongguan, China), as shown in Figure 2. The testing machine has a maximum loading capacity of 1 kN. During the tests, the specimens were clamped at the two ends by grippers for a length of 16 mm. The tensile velocity was set to 500 mm/min. The variation in tensile force (*F*) with tensile distance (*d*) was recorded. The tensile distance *d* was directly measured as the moving distance of the upper gripper, which is the total deformation of the sample. Tension tests were conducted for three specimens. The tests for the three specimens are denoted as tests 1, 2, and 3 in the following. The tensile force–distance (*F*–*d*) curves measured from the tests are given in Figure 3. The curves agree well with each other. For each sample, the deformation at fracture can reach more than three times the length of the middle part of the sample where deformation mainly occurs. This indicates that the polyurea possesses good ductility.

### 3.2. Inverse Parameter Identification

An inverse method is presented for the identification of material parameters in the hyperelastic constitutive model for characterizing the tensile performance of the polyurea. In the method, a finite element (FE) model for the uniaxial tension test is established based on the finite element software ABAQUS [22], and an iterative strategy is proposed to optimize the material parameters to seek a good match between the tensile force–distance curves obtained from the FE model and the test.

The FE model is shown in Figure 4. The model is established for test 1. The actual geometry of the specimen is used in the model. The actual dimensions of the specimen show some differences to the nominal ones given in Table 1 due to contraction in the drying process. In order to generate the actual geometry of the specimen, contraction ratios in length and width were measured. They were determined as the ratios of the actual total length and width of the middle part to the corresponding nominal ones, which were 0.938 and 0.892, respectively. The in-plane geometry of the specimen was constructed by scaling the geometry in Figure 1a with the contraction ratios determined above. The thickness of the specimen in the model was taken as that measured from the specimen (1.8 mm). The specimen was discretized with 40,064 elements. The surface nodes of the lower end of the specimen with a length of 16 mm were constrained in the *x*-, *y*-, and *z*-directions. The surface nodes of the upper end of the specimen with a length of 16 mm were given the actual tensile velocity in the *y*-direction and constrained in the *x*- and *z*-directions.

In the iterative strategy for the identification of material parameters, initial values of these parameters should be estimated first. To this end, the principal stretch ratio at tensile distance *d* is estimated as
(12)$$\lambda_{U}^{0}\left(d\right)=d/l_{0},$$
and the nominal stress at tensile distance *d* is estimated as
(13)$$T_{U}^{0}\left(d\right)=F\left(d\right)/S_{0},$$
where *l*_0_ is the actual length of the middle part of the specimen, which is 30.97 mm, $F\left(d\right)$ is the tensile force at tensile distance *d* obtained from the test, and *S*_0_ is the actual initial area of the cross-section of the specimen, which is 9.63 mm^2^. Based on Equations (12) and (13), the *T*_U_–*λ*_U_ curve is estimated. Using the least-squares method, we fit the curve with Equation (11) and obtained the initial values of material parameters (see the material parameters for iteration number 0 in Table 2).

The initial material parameters were used in the FE model to simulate the uniaxial tension test. Figure 5 shows the simulated results with these parameters. As seen from Figure 5a, the stress distribution on the cross-section of the middle part of the specimen is quite uniform. Thus, the nominal stress of the middle part estimated with Equation (12) should be accurate. From Figure 5b, it can be seen that the middle part of the specimen deformed uniformly. However, beside the middle part, the other part of the specimen also shows obvious deformation and contributes to the total deformation of the specimen. Figure 6a gives a comparison between the total deformation of the specimen (*d*) and the deformation of the middle part (Δ*l*), and Figure 6b gives the deformation ratio *θ* (*θ* = *d*/Δ*l*). It can be clearly seen that there are significant differences between the two deformations, so estimating the principal stretch ratio of the middle part with Equation (13) will result in significant error, and generally, this will produce larger principal stretch ratios than the true ones. Thus, the material parameters obtained based on $T_{U}^{0}$ and $\lambda_{U}^{0}$ are inaccurate and need to be corrected.

In this work, the material parameters were iteratively corrected until the *F*-*d* curves obtained from the test and the FE simulation were in a good agreement. The iterative correction process is given as follows:
(1)The deformation ratio for the *n*th iteration ($\theta^{n}$) is calculated with the *n*th FE simulation result. It is calculated for *m* tensile distances where FE simulation results are output. The deformation ratios at different tensile distances $\theta^{n}\left(d_{i}\right)$ (*i* = 1, 2, …, *m*) are fitted with the following 7th-order polynomial
(14)$$\theta^{n}\left(d\right)=a_{7}d^{7}+a_{6}d^{6}+a_{5}d^{5}+a_{4}d^{4}+a_{3}d^{3}+a_{2}d^{2}+a_{1}d^{1}+a_{0}.$$(2)The principal stretch ratio of the middle part for the (*n* + 1)th iteration is corrected as
(15)$$\lambda_{U}^{n+1}\left(d\right)=\lambda_{U}^{0}\left(d\right)/\theta^{n}\left(d\right).$$The corrected principal stretch ratio $\lambda_{U}^{n+1}\left(d\right)$ is calculated at *k* tensile distances, which are recorded in the tensile test. The nominal stresses at the same *k* tensile distances are calculated with Equation (13). They remain the same as the initial values and require no correction. The reason for this is that nominal stress can be estimated by Equation (13) with very good accuracy, as stress distribution on the cross-section of the specimen is quite uniform, as seen in Figure 5a. With the above calculations, the $T_{U}^{0}\left(d\right)$-$\lambda_{U}^{n+1}\left(d\right)$ curve is constructed.(3)The least-squares method is used to fit the $T_{U}^{0}\left(d\right)$-$\lambda_{U}^{n+1}\left(d\right)$ curve with Equation (11), and the material parameters for the (*n* + 1)th iteration is obtained.(4)The (*n* + 1)th FE simulation is conducted with the material parameters obtained in Step (3).(5)The *F*-*d* curve is output for the (*n* + 1)th FE simulation and compared with the one measured from the test. If *F*-*d* curves obtained from the test and the FE simulation achieves a good agreement, then the iterative process is terminated. If not, go to Step (1).

The convergence of the above iterative process is very fast. Three iterations are only required to be performed. Figure 7 gives the convergence of deformation ratio θ and tensile force *F*. Table 3 gives the coefficients of the 7th-order polynomial deformation ratio function for different iterations. Table 1 gives the material parameters for different iterations. As seen from Figure 7b, the *F*-*d* curves obtained from the test and the FE simulation achieve an excellent agreement after three iterations. Thus, the material parameters obtained from the third iteration were taken as the final result and used to characterize the tensile performance of the polyurea.

## 4. Inverse Identification of Material Parameters for Compressive Performance

### 4.1. Uniaxial Compression Test

Uniaxial compression tests were performed to provide experimental data to inversely identify material parameters for the compressive performance of the polyurea. The specimens used in the tests were cylindrical. The nominal diameter and height of the cylindrical specimen were 29.0 and 12.5 mm, respectively. Similar to the tensile specimens, the compressive specimens were fabricated by the casting method with the mold shown in Figure 8a. A typical specimen is shown in Figure 8b. Three compressive specimens were used in the following tests. The actual dimensions of these specimens are listed in Table 4. They were slightly smaller than the nominal ones, mainly due to contraction in the drying process.

Uniaxial compression tests were performed with a universal testing machine produced by Shanghai Hualong Test Instruments Co., Ltd. (Shanghai, China), as shown in Figure 9. The testing machine has a maximum loading capacity of 100 kN. The compressive velocity was set to 150 mm/min, which makes the strain rate of compression tests comparable to that of the tension tests. The total compression distance was set to 5.6 mm. In the compression test, it is important to reduce the effect of friction, so lubricants are usually used [23]. In this work, silicone oil was used as lubricant to reduce the friction between contact surfaces of the specimen and the loading device. The variation in compressive force (*F*) with compressive distance (*d*) was recorded for three specimens by three tests, which are denoted as tests 1, 2, and 3 in the following. The compressive force–distance curves measured from the tests are given in Figure 10. Overall, the test results for the three specimens are in good agreement, though some differences can be observed. Figure 11 shows the specimens at the maximum compressive distance. It can be seen that the samples become drum-shaped. This is mainly because frictional forces exist at the top and bottom surfaces and resist the lateral deformations in the samples, which results in a deviation for the ideal uniaxial stress state.

### 4.2. Inverse Parameter Identification

The stress and strain distributions in the compressive specimens are not as uniform as those in the middle parts of the tensile specimens, so the inverse method for identifying material parameters for tensile performance is not applicable to the identification of material parameters for compressive performance. Here, an inverse method based on the response surface is presented to identify material parameters in the hyperelastic constitutive model for characterizing the compressive performance of the polyurea. The compressive *F*–*d* curve of test 1 was used for inverse identification.

The procedure for the response surface-based inverse method is given in the following description. First, initial material parameters are estimated with the compressive *F*–*d* curve from the test. Secondly, a sensitivity analysis of material parameters is performed to determine the material parameters being identified and their search ranges, after which the Latin hypercube sampling (LHS) method is employed to generate samples. Then, an FE model of the uniaxial compression test is established. With this model, simulated compressive *F*–*d* curves are obtained for all the samples. Then, errors between compressive forces obtained from FE simulations and the test are calculated, and an RBF response surface model is constructed for the error of compressive force. Finally, the genetic algorithm is employed to minimize the error of the compressive force and obtain optimal material parameters.

The initial material parameters for characterizing the compressive performance were estimated in a similar way in Section 3.1. Based on the assumption of an ideal uniaxial stress state for the compressive specimen, the principal stretch ratio at compressive distance *d* is estimated as
(16)$$\lambda_{U}^{}\left(d\right)=\left(l_{0}^{\prime }-d\right)/l_{0}^{\prime },$$
and the nominal stress at compressive distance *d* is estimated as
(17)$$T_{U}^{}\left(d\right)=F\left(d\right)/S_{0}^{\prime },$$
where $l_{0}^{\prime }$ is the actual initial height of the compressive specimen, and $S_{0}^{\prime }$ is the actual initial cross-sectional area of the specimen. For test 1, $l_{0}^{\prime }$ and $S_{0}^{\prime }$ are 11.17 mm and 552.93 mm^2^, respectively. The *T*_U_-*λ*_U_ relationship estimated by Equations (16) and (17) is fitted with Equation (11) by the least-squares method. The initial material parameters are obtained and given in Table 5. These material parameters are not accurate, because the actual stress state is not a strictly uniaxial stress state, due to the existence of friction, which can be seen in the FE simulation result of the compression test with these parameters (see Figure 12). However, the stress–strain curve defined by these parameters will be approximate to the true one or in the same order at least. Thus, they were used as baseline values below for determining parameters’ search ranges in the inverse identification.

A sensitivity analysis was performed to observe the effects of material parameters on the uniaxial stress–strain response defined by the second-order polynomial hyperelastic model. Each parameter was analyzed independently and ranged from $\left(1-\omega \right)C_{ij}^{0}$ to $\left(1+\omega \right)C_{ij}^{0}$, where *ω* is a factor. *ω* was set to 2% in the sensitivity analysis. Figure 13 shows the sensitivity of the uniaxial stress–strain response to different material parameters. The stress–strain response is sensitive to all the five material parameters. Thus, all of these parameters are taken as parameters required to be identified. From Figure 13, it can be seen that unreasonable stress–strain responses exist in some cases, for example, stress becomes positive at a principal stretch ratio of 0.4 when *C*_20_ is $0.98C_{20}^{0}$, which means tensile stress occurs at large compressive strain. In the following parameter identification, to avoid this unreasonable stress–strain response in the samples, search ranges of material parameters were determined as those given in Table 4, namely, *ω* was taken as 1.7%, 1.7%, 1.7%, 0.5%, and 0.5% for $C_{10}^{0}$, $C_{01}^{0}$, $C_{20}^{0}$, $C_{11}^{0}$, and $C_{02}^{0}$, respectively.

To provide samples for constructing a surface response model, the LHS method was employed, and 50 samples were generated, as shown in Table A1 (see the Appendix A). For each sample, an FE simulation was carried out with the material parameters of each sample to produce compressive forces during the compressive process. The FE model used for the simulation of the compression test is shown in Figure 14. The two load cells were modeled with rigid surfaces. The specimen was discretized with 51,678 elements. The lower load cell was fixed, and the upper load cell was given the actual compressive velocity in the *y*-direction. A friction coefficient of 0.19, whose determination is detailed below, was set for the contact between the load cells and the specimen.

The friction coefficient was experimentally determined as shown in Figure 15. The specimen was placed on the load cell of the compression test machine. A weight of 250 g was placed on the upper surface of the specimen to increase the frictional force and reduce the test error. The lubrication condition between the specimen and the load cell was set as same as that in the compression test. A digital force gauge with a measuring range of 0 to 2 N was driven slowly by a guide screw under the control of a digital motion controller to pull the specimen horizontally and slowly. The force gauge recorded the maximum frictional force during the process of the specimen from the static state to the moving state. The friction coefficient was calculated with the maximum force and the weight of the specimen. Three specimens were tested. The friction test was repeated eight times for each specimen. The average friction coefficient of the three specimens was 0.19.

For each sample, an error between the compressive forces obtained from the FE simulation and the test was calculated. The error is defined as
(18)$$e_{F}\left(x_{i}\right)=\sum_{j=1}^{N}{\left[F_{S}^{}\left(x_{i},\;d_{j}\right)-F_{T}^{}\left(d_{j}\right)\right]}^{2}/F_{T}^{}{}^{2}\left(d_{j}\right),$$
where $x=\left[C_{10}\;C_{01}\;C_{20}\;C_{11}\;C_{02}\right]$ is the parameter vector, *F*_S_ denotes the compressive force obtained from the FE simulation, *F*_T_ is the compressive force obtained from the test, *d_i_* is the *i*th compressive distance at which compressive force is reported, and *N* is the number of reporting points of compressive force.

With the samples in Table A1, an RBF surface response model for the error of compressive force was constructed. The form of the RBF surface response model is:(19)$$e_{F}\left(x\right)=\sum_{i=1}^{M}w_{i}\varphi \left(r_{i}\right),$$
where *M* is the number of samples that is 50, as mentioned above, *w* is the weight, *φ* is the radial basis function, and $r_{i}=\left\|x-x_{i}\right\|/10^{6}$ is the Euclid distance between an arbitrary point and the sample *i*. In this work, *φ* was chosen as the Gaussian function whose expression is
(20)$$\varphi \left(r\right)=e^{-\alpha r^{2}},$$
where *α* is a constant, which is usually taken as 1/*η*, where *η* is the number of input variables (material parameters) for a sample. *η* was 5 in this work. The weights for the RBF surface response model are given in Table 6.

To validate the accuracy of the RBF surface response model, thirty random samples were generated. Figure 16 compares the errors of compressive force obtained with the surface response model and FEM simulation. It can be seen that the surface response model produces good accuracy.

To identify the material parameters, the GA algorithm was employed to minimize the error function of compressive force. The population size was set to 100, and the maximum generation was set to 200, when using the GA algorithm. The identified material parameters are given in Table 5. With these material parameters, an FE simulation of the compressive test was performed. Figure 17 compares the compressive forces obtained from the simulation and the test. The two compressive curves agree well with each other. This proves the accuracy of the material parameters and confirms the effectiveness of the above parameter identification method.

## 5. Material Parameters Characterizing Both Tensile and Compressive Performance

In Section 4 and Section 5, two sets of material parameters are separately identified. They are accurate for describing tension and compression performance, respectively. For convenience, the two sets of parameters are referred to as tension and compression parameters, respectively. Figure 18 shows the uniaxial stress–strain curves defined by the tension and compression parameters for a wide strain range. As seen in the figure, the tension parameters cannot be extended to accurately characterize the compression performance and vice versa.

To overcome the above problem, a complete identified stress–strain curve (shortened to identified curve for the following description) was formed by combining the tensile part of the uniaxial stress–strain curve defined by the tension parameters and the compressive part of uniaxial stress–strain curve defined by the compression parameters. This identified curve accurately describes both the tension and compression behaviors of the polyurea. The curve was fitted with different-order polynomial hyperelastic constitutive models through the least-squares method. Table 7 gives the fitted material parameters, and Figure 19 shows the uniaxial stress-strain curves defined by the fitted parameters (shortened to fitted curves for the following description).

As seen from Figure 18 and Figure 19, the three sets of fitted parameters based on the identified curve appropriately describe both uniaxial tension and compression behaviors of the polyurea and produce a much better description than the tension parameters and compression parameters. Among the three models, the third-order model gives the best description for the stress–strain behavior of the polyurea. Using different-order hyperelastic constitutive models with the fitted parameters in Table 7, we performed simulations of the tension and compression tests. Figure 20 gives a comparison between the tensile and compressive forces obtained from the simulations and the corresponding tests. For the tensile test, the simulated results of different-order models achieve a very good agreement with the experimental results. The L2 relative errors between the simulated and experimental results are 5.2%, 3.9%, and 1.5% for the first-, second-, and third-order models, respectively. For the compression test, the simulated results also qualitatively agree well with the experimental result, but relatively large errors are observed. The L2 relative errors are 23.7%, 22.4%, and 15.7%, respectively. According to the above analysis, the third-order model is the best choice for the polyurea in terms of accuracy.

## 6. Conclusions

An inverse procedure was developed to identify material parameters of a hyperelastic model of polyurea materials. With this procedure, material parameters of a one-part polyurea SWD9526 were obtained. The material parameters for tensile performance were obtained with an iterative inverse method, and the material parameters for compressive performance were determined with a response surface-based inverse method. The two sets of parameters can describe well the tensile and compressive behaviors of the polyurea, respectively. Through the examination of material behaviors in a large strain range, it was found that tensile parameters cannot be extended to describe compressive behaviors and vice versa. This implies that we should be cautious when the hyperelastic model parameters obtained from a given loading condition are extended to general conditions. Attempts were made to determine the material parameters, which are accurate for describing both tensile and compressive performance of the polyurea. Different-order polynomial hyperelastic models were employed. Parameters for the first-, second-, and third-order polynomial models were obtained by fitting a uniaxial stress–strain curve that includes both tension and compression deformations. The parameters appropriately characterize both tensile and compressive behaviors of the polyurea. With these parameters, FE simulations of tension and compression tests were performed. The simulated results showed that the third-order model produces the best accuracy. The inverse procedure presented in this work can simplify the tensile test procedure and consider the effect of friction in the compression test. The constitutive parameters of polyurea SWD9526 obtained here is useful for the design of engineering structures with the polyurea.

## Figures and Tables

**Figure 1 polymers-13-02253-f001:**
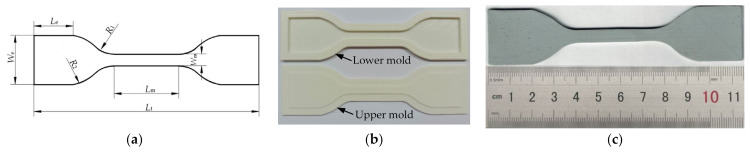
Specimen and its fabrication: (**a**) shape of specimen; (**b**) mold; (**c**) practical specimen.

**Figure 2 polymers-13-02253-f002:**
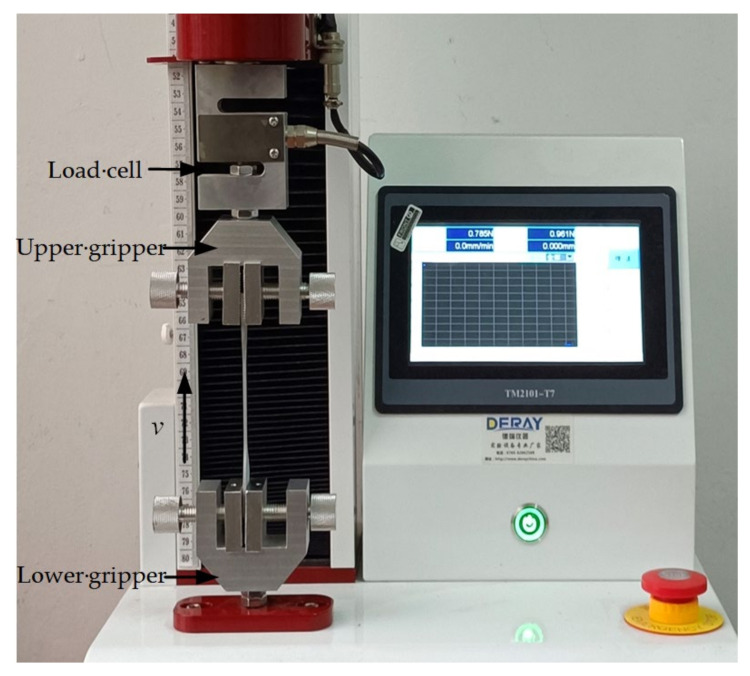
Experimental configuration for tension test of polyurea.

**Figure 3 polymers-13-02253-f003:**
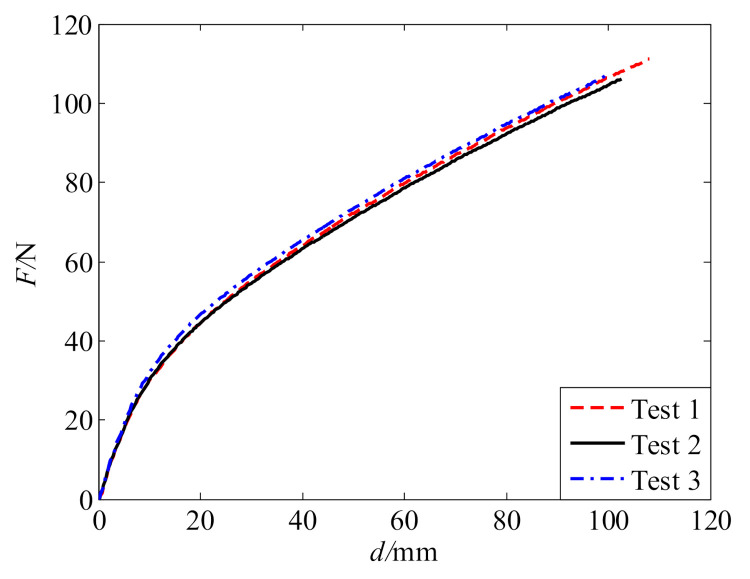
Tensile force–distance curves.

**Figure 4 polymers-13-02253-f004:**
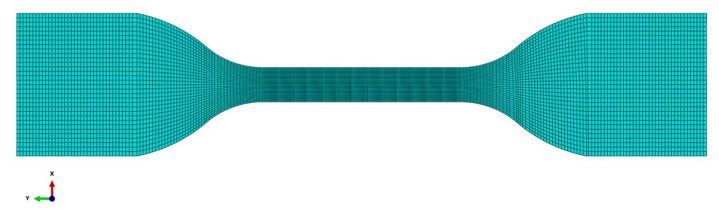
FE model for tension test of polyurea.

**Figure 5 polymers-13-02253-f005:**
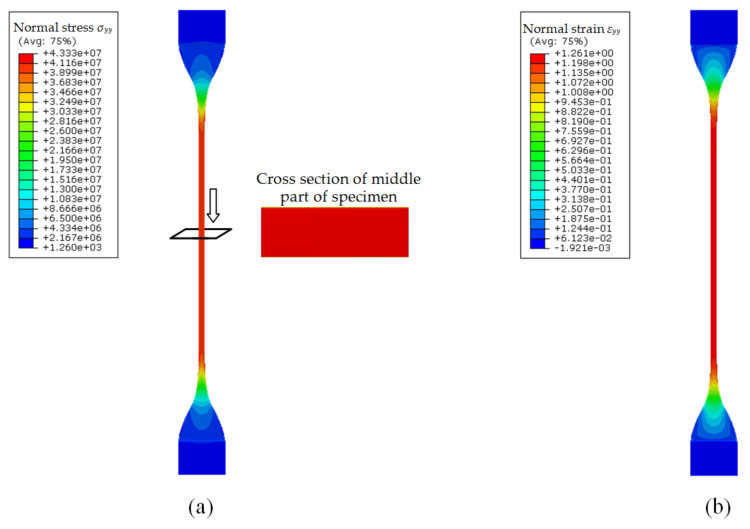
FE simulation results of uniaxial tension test: (**a**) true stress in the loading direction; (**b**) true strain in the loading direction.

**Figure 6 polymers-13-02253-f006:**
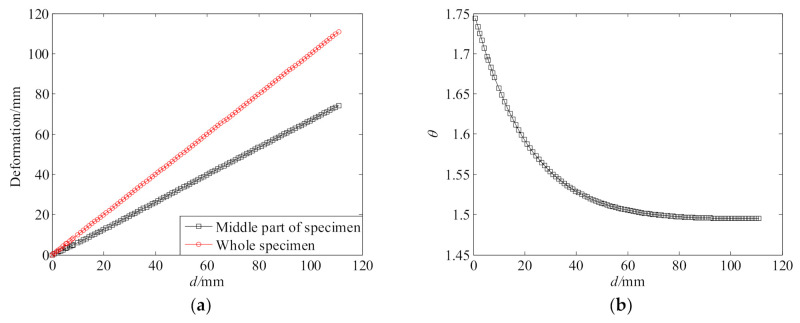
Deformations of different parts of the specimen: (**a**) total deformation and middle part’s deformation; (**b**) ratio of the two deformations.

**Figure 7 polymers-13-02253-f007:**
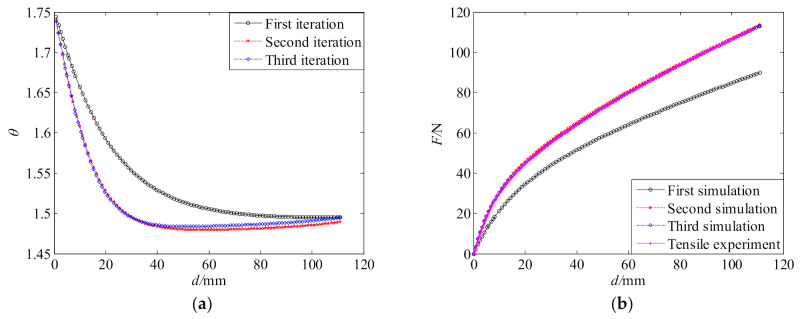
Convergence: (**a**) deformation ratio; (**b**) tensile force.

**Figure 8 polymers-13-02253-f008:**
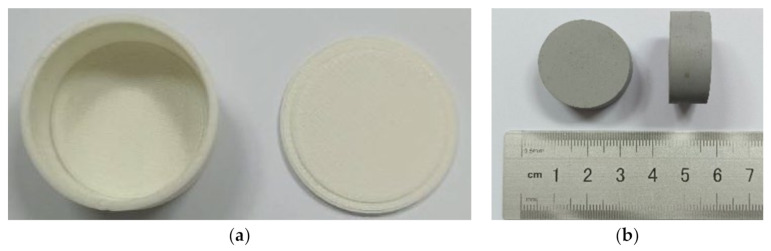
Fabrication of specimen: (**a**) mold for fabricating specimen; (**b**) specimen.

**Figure 9 polymers-13-02253-f009:**
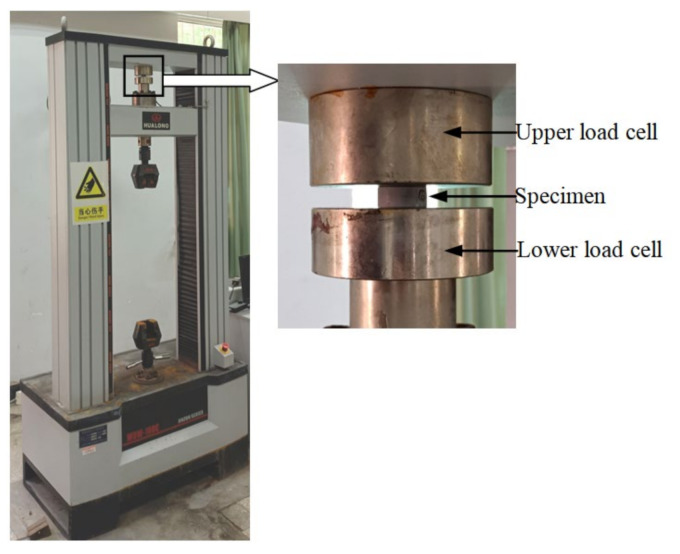
Experimental configuration for compression test.

**Figure 10 polymers-13-02253-f010:**
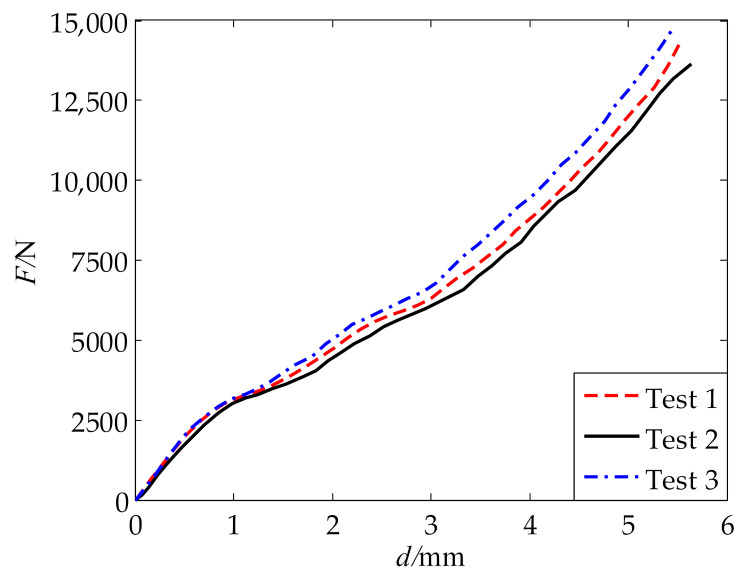
Compressive force–distance curves.

**Figure 11 polymers-13-02253-f011:**
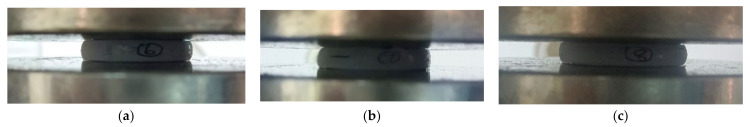
Deformations of specimens at the maximum compressive distance: (**a**) test 1; (**b**) test 2; (**c**) test 3.

**Figure 12 polymers-13-02253-f012:**
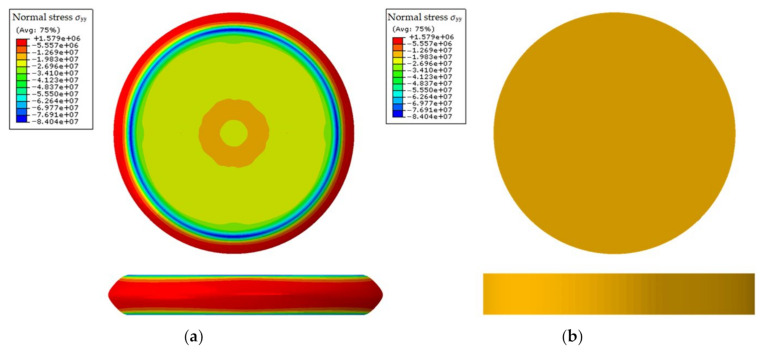
Stress distributions from FE simulation of uniaxial compression test with initial material parameters: (**a**) using actual friction coefficient (0.19); (**b**) no friction.

**Figure 13 polymers-13-02253-f013:**
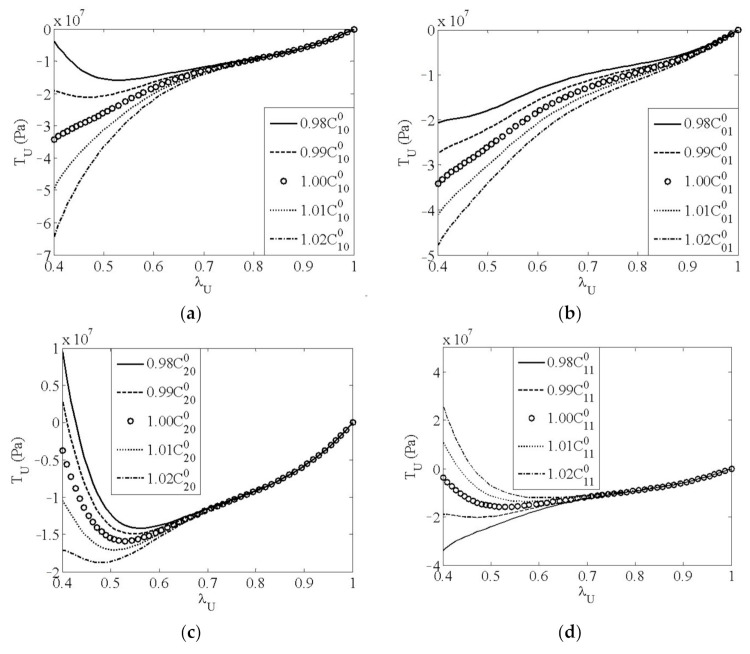
Sensitivity of uniaxial stress–strain response to material parameters: (**a**) *C*_10_; (**b**) *C*_01_; (**c**) *C*_20_; (**d**) *C*_11_; (**e**) *C*_02_.

**Figure 14 polymers-13-02253-f014:**
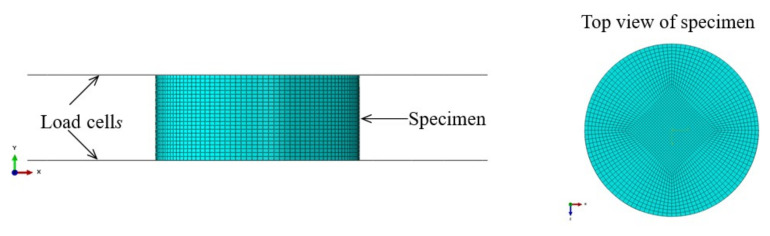
FE model for compression test.

**Figure 15 polymers-13-02253-f015:**
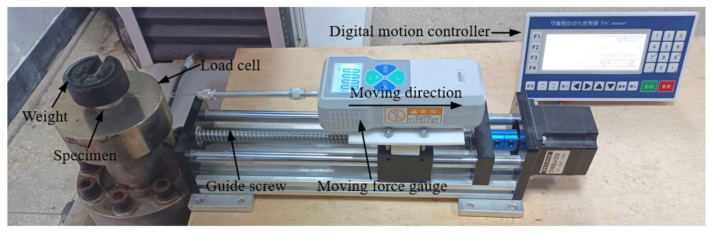
Friction coefficient test configuration.

**Figure 16 polymers-13-02253-f016:**
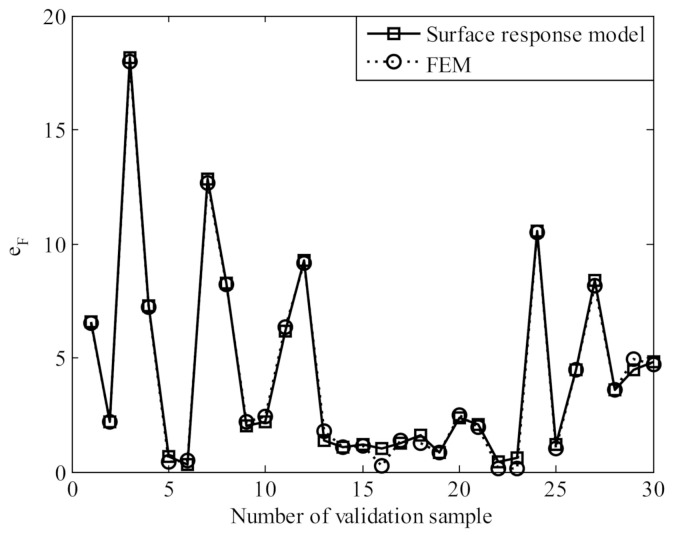
Validation of surface response model.

**Figure 17 polymers-13-02253-f017:**
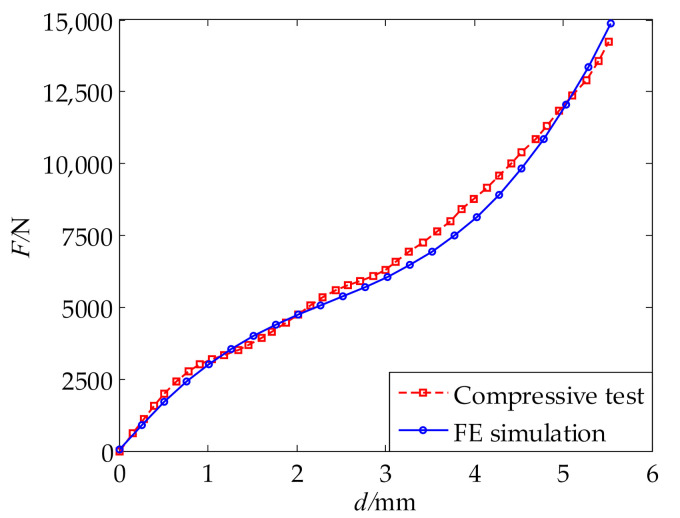
Comparison between test result and simulated result with the identified parameters.

**Figure 18 polymers-13-02253-f018:**
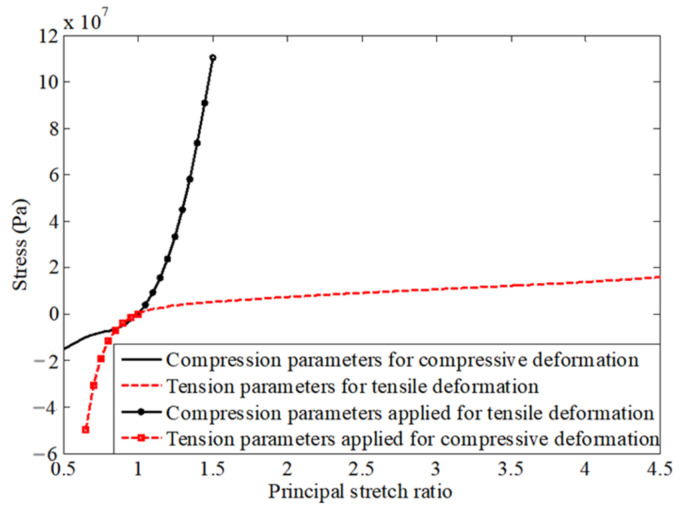
Uniaxial stress-strain curves over a wide strain range for two sets of material parameters.

**Figure 19 polymers-13-02253-f019:**
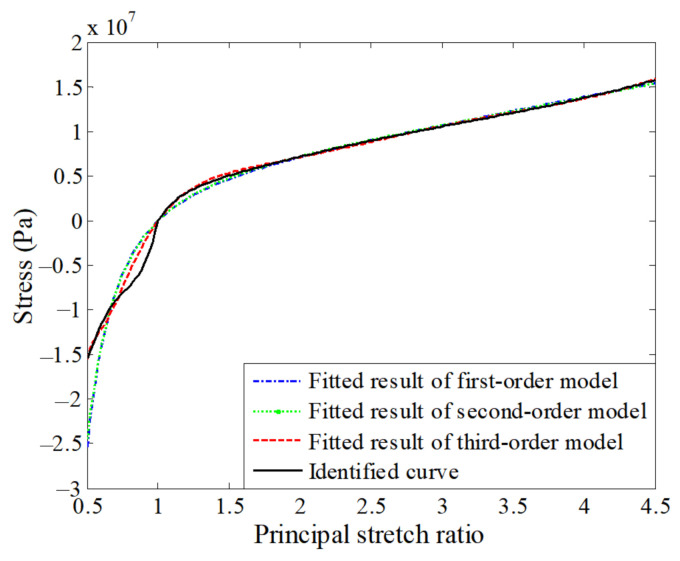
Fitted result of polynomial hyperelastic model.

**Figure 20 polymers-13-02253-f020:**
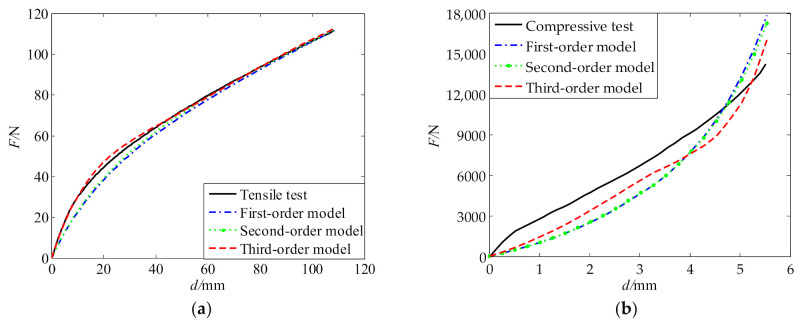
Comparison between simulated results with different-order models and experimental results for different material tests: (**a**) tension test; (**b**) compression test.

**Table 1 polymers-13-02253-t001:** Nominal dimensions of specimen.

Name	Symbol	Value/mm
Total length	*L* _t_	115
Width of two ends	*W* _e_	25
Length of two ends	*L* _e_	20
Length of middle part	*L* _m_	33
Width of middle part	*W* _m_	6
Radius of concave transition edge	*R* _1_	14
Radius of convex transition edge	*R* _2_	25

**Table 2 polymers-13-02253-t002:** Material parameters for tensile performance of polyurea.

Iteration Number	*C*_10_/Pa	*C*_01_/Pa	*C*_20_/Pa	*C*_11_/Pa	*C*_02_/Pa
0	−186,300	2,843,500	−2900	1200	351,300
1	−5,035,000	9,433,700	63,700	−419,400	2,462,500
2	−5,368,900	9,743,600	72,800	−476,300	2,657,200
3	−5,413,200	9,787,600	75,200	−490,700	2,695,200

**Table 3 polymers-13-02253-t003:** Coefficients of polynomial deformation ratio function.

Iteration Number	*a* _7_	*a* _6_	*a* _5_	*a* _4_	*a* _3_	*a* _2_	*a* _1_	*a* _0_
1	5.1822 × 10^6^	−2.0426 × 10^6^	2.8397 × 10^5^	−1.0173 × 10^4^	−1.5652 × 10^3^	218.1200	−11.7220	1.7534
2	3.2930 × 10^7^	−1.2729 × 10^7^	1.7989 × 10^6^	−9.4296 × 10^5^	−2.0048 × 10^3^	451.9100	−19.0830	1.7560
3	3.7495 × 10^7^	−1.4333 × 10^7^	1.9980 × 10^6^	−1.0239 × 10^5^	−2.3050 × 10^3^	485.8700	−19.7710	1.7564

**Table 4 polymers-13-02253-t004:** Actual dimensions of specimens.

Specimen Number	Diameter/mm	Height/mm	Weight/g
1	26.54	11.17	7.99
2	26.40	11.20	7.92
3	26.37	11.15	8.06

**Table 5 polymers-13-02253-t005:** Initial values, search range, and identified values of material parameters for comprehensive performance.

Parameter	Initial Value/Pa	Search Range/Pa	Identified Value/Pa
*C* _10_	57,909,000	[56,924,547, 58,893,453]	57,462,445
*C* _01_	−45,375,000	[−46,146,375, −44,603,625]	−46,146,375
*C* _20_	30,132,000	[29,619,756, 30,644,244]	30,155,937
*C* _11_	−13,617,000	[−13,685,085, −13,548,915]	−13,548,915
*C* _02_	2,813,000	[2,798,935, 2,827,065]	2,827,065

**Table 6 polymers-13-02253-t006:** Weights for surface response model.

Weight	Value	Weight	Value	Weight	Value	Weight	Value	Weight	Value
*w* _1_	−19,324.12	*w* _11_	13,911.72	*w* _21_	14,859.58	*w* _31_	7542.22	*w* _41_	13,704.27
*w* _2_	−5524.97	*w* _12_	14,329.52	*w* _22_	39,776.94	*w* _32_	−38,263.53	*w* _42_	−15,283.83
*w* _3_	−8940.66	*w* _13_	2048.72	*w* _23_	−30,647.00	*w* _33_	19,663.34	*w* _43_	143.39
*w* _4_	18,555.15	*w* _14_	32,648.17	*w* _24_	−3056.50	*w* _34_	14,514.78	*w* _44_	49,202.01
*w* _5_	3652.96	*w* _15_	−2106.49	*w* _25_	17,048.97	*w* _35_	−7536.94	*w* _45_	−4307.05
*w* _6_	13,630.87	*w* _16_	3199.16	*w* _26_	−49,268.38	*w* _35_	1701.26	*w* _46_	1662.73
*w* _7_	561.36	*w* _17_	3161.56	*w* _27_	11,733.29	*w* _37_	−1693.64	*w* _47_	−33,476.46
*w* _8_	42,337.42	*w* _18_	−13,972.10	*w* _28_	1604.59	*w* _38_	−3278.22	*w* _48_	−33,254.66
*w* _9_	4363.38	*w* _19_	−9061.07	*w* _29_	−34,785.08	*w* _39_	−26,533.68	*w* _49_	−10,181.13
*w* _10_	−43,354.89	*w* _20_	7439.65	*w* _30_	25,865.51	*w* _40_	21,541.27	*w* _50_	−6418.09

**Table 7 polymers-13-02253-t007:** Material parameters of polynomial hyperelastic constitutive models.

Model Order	*C*_10_/Pa	*C*_01_/Pa	*C*_20_/Pa	*C*_11_/Pa	*C*_02_/Pa	*C*_30_/Pa	*C*_21_/Pa	*C*_12_/Pa	*C*_03_/Pa
First	1,492,265	1,069,365	/	/	/	/	/	/	/
Second	1,584,196	1,036,727	5038	−24,416	−7969	/	/	/	/
Third	1,639,722	1,980,931	−10,743,503	21,625,125	−11,805,576	−4689	88,598	2,213,877	−461,844

## Data Availability

All data presented in this study are available upon request.

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
