# Peer review of "Inverse Parameter Identification for Hyperelastic Model of a Polyurea"

_polymers, 2021, doi:10.3390/polym13142253_

Round 1

Reviewer 1 Report

Polyurea is a flexible material which has a wide application. In this paper, an inverse procedure is proposed to identify material parameters of polyurea materials. It seems to improve the accuracy and reliability. This paper is interesting and has a significance for the flexible polymer community. Before accepted, some issues should be addressed, as below.

  1. This work illustrated that “the present inverse procedure can simplify experimental configurations and consider effects of friction in compression test”. The friction has an important effect on the deformation of flexible polymers upon compression. Some papers have given, such as Polymer Testing 70 (2018) 192-196. Could the authors give a detailed illustration on how to determine the friction parameter for modeling which can link the experimental case.
  2. Figure 12 gives the stress distributions from FE simulation of uniaxial compression test with initial material parameters: (a) using actual friction coefficient (0.19); (b) no friction. I doubt (a) and (b) are inverse? Please double check and give more illustrations.
  3. Could the authors give the chemical composition of the studied Polyurea.
  4. It is better to give a schematic drawn picture for illustrating the constitutive model.
  5. The references are too less. Currently, flexible polymers and polymer composites have attracted an extensive attention, such as Journal of Materials Science and Technology 57 (2020) 12-17, Polymers 11 (2019) 467-478, Composites Part B 152 (2018) 96-101, Material & Design 79 (2015) 73-85, Polymer 65 (2015) 72-80, and so on.
  6. The Figures are better to be improved for more beautiful.

Reviewer 2 Report

Dear Authors!

It was interesting to me to review your manuscript containing such a thorough technical and mathematical study. In terms of research content and novelty, the article can be published as-is. However, I will provide a few recommendations that can help improve the reader's perception of the results presented.

1. I recommend using the notation with indices in Fig. 1 and in the table instead of A, B, C ... For example, Length of middle part - Lm.

2. Figure 1 (a) should be enlarged to improve readability.

3. Designations from the legend fig. 3 and 10 must be entered in the text.

4. Legend fig. 7, (a) is absolutely uninformative.

5. Table 6 can be moved to applications.

Moreover, the article must be read by a native speaker.

Round 2

Reviewer 1 Report

This paper can be accepted in present form.